# Identifying Obstructive Sleep Apnoea in Patients with Empty Nose Syndrome

**DOI:** 10.3390/diagnostics12071720

**Published:** 2022-07-15

**Authors:** Chien-Chia Huang, Pei-Wen Wu, Chi-Cheng Chuang, Cheng-Chi Lee, Yun-Shien Lee, Po-Hung Chang, Chia-Hsiang Fu, Chi-Che Huang, Ta-Jen Lee

**Affiliations:** 1Division of Rhinology, Department of Otolaryngology, Chang Gung Memorial Hospital, Taoyuan 33382, Taiwan; agar@cgmh.org.tw (C.-C.H.); a9665@cgmh.org.tw (P.-W.W.); bc1766@gmail.com (P.-H.C.); fufamily@cgmh.org.tw (C.-H.F.); 2Graduate Institute of Clinical Medical Sciences, College of Medicine, Chang Gung University, Taoyuan 33301, Taiwan; 3School of Medicine, Chang Gung University, Taoyuan 33301, Taiwan; 4Department of Neurosurgery, Chang Gung Memorial Hospital, Taoyuan 33382, Taiwan; ccc2915@cgmh.org.tw (C.-C.C.); yumex86@gmail.com (C.-C.L.); 5Department of Biomedical Engineering, National Taiwan University, Taipei 10617, Taiwan; 6Genomic Medicine Research Core Laboratory, Chang Gung Memorial Hospital, Taoyuan 33382, Taiwan; bojack@mail.mcu.edu.tw; 7Department of Biotechnology, Ming Chuan University, Taoyuan 11103, Taiwan; 8Department of Otolaryngology, Xiamen Chang Gung Hospital, Xiamen 361028, China

**Keywords:** empty nose syndrome 6-item questionnaire, empty nose syndrome, obstructive sleep apnoea, polysomnography, sleep quality

## Abstract

Obstructive sleep apnoea (OSA) is characterised by repetitive episodes of upper airway collapse and breathing cessation during sleep. Empty nose syndrome (ENS) is a surgically iatrogenic phenomenon of paradoxical nasal obstruction despite an objectively patent nasal airway. This study aimed to investigate sleep quality and the presence of OSA in ENS patients. Forty-eight ENS patients underwent full-night polysomnography. Total nasal resistance (TNR) was determined using anterior rhinomanometry. Symptoms and quality of life were evaluated by the empty nose syndrome 6-item questionnaire (ENS6Q), Sino-Nasal Outcome Test-22 (SNOT-22), and Epworth Sleepiness Scale questionnaires (ESS). Fourteen, twelve, and fourteen patients had mild, moderate, and severe OSA, respectively. The apnoea–hypopnoea index (AHI) and the lowest SpO_2_ were 23.8 ± 22.4/h and 85.9 ± 11.1%, respectively. N1, N2, N3 and rapid-eye-movement sleep comprised 30.2 ± 16.9%, 47.3 ± 15.5%, 2.1 ± 5.4%, and 20.0 ± 8.1% of the total sleep time. Body mass index, neck circumference, serum total immunoglobulin E, and ENS6Q score were significantly associated with AHI in the regression analysis. The ENS6Q scores correlated positively with AHI, arousal index, and ESS score, but negatively with TNR. ENS patients showed a high OSA prevalence and significant sleep impairment. The extent of OSA was associated with obesity levels and ENS symptom severity. The ENS6Q scores correlated negatively with nasal resistance, and positively with arousal frequency and daytime sleepiness. The recognition of individuals experiencing marked OSA and provision of appropriate intervention is critical to preventing long-term morbidity and mortality, and improving therapeutic outcomes in ENS patients.

## 1. Introduction

Obstructive sleep apnoea (OSA) is a complex disorder characterised by repetitive episodes of upper airway collapse and breathing cessation during sleep [1]. The drop in muscle tone during sleep leads to collapse of the upper airways and intermittent episodes of hypopnoea (partial obstruction) and/or apnoea (total obstruction). OSA not only causes symptoms such as mid-night arousal and excessive daytime sleepiness, but is also associated with a significant risk of cardiovascular, pulmonary, and neurocognitive morbidity and mortality [2].

Empty nose syndrome (ENS) is a surgically iatrogenic phenomenon of paradoxical nasal obstruction despite an objectively patent nasal airway [3]. The pathophysiology of abnormal sensations in ENS patients is complicated and not fully understood [4]. Hypothesised theories include changes in nasal airflow after surgical intervention and abnormalities in neurosensory systems owing to improper mucosal healing [4,5,6,7,8].

ENS patients may also experience dyspnoea, nasal and pharyngeal dryness, facial or nasal pain, rhinorrhoea or postnasal drip, crusting, hyposmia, headache, inability to concentrate, chronic fatigue, frustration, anxiety, and depression [1,9,10]. Affected patients often report significant psychological effects, loss of productivity, decreased quality of life (QOL), and lifestyle disruption [11,12,13]. Our previous questionnaire-based study revealed associations between significant sleep dysfunction and psychological effects in ENS patients [14,15]. However, there is no study that objectively measures sleep in the literature. As ENS affects the upper airway and exhibits significant sleep dysfunction, it is necessary to conduct an objective study to investigate sleep quality and the presence of OSA in these patients. As a result, we firstly used polysomnography (PSG), the gold standard for the evaluation of sleep architecture and diagnosis of OSA, in patients with ENS and identified the clinical risk factors for moderate-to-severe OSA in this study. This information could be helpful for sleep dysfunction management and optimisation of patient-centred care.

## 2. Materials and Methods

### 2.1. Patients

After approval from the institutional review board of Chang Gung Medical Foundation (IRB numbers: 201702048B0, 201802147A3 and 201902001A3), we recruited ENS patients who were managed in the Department of Otolaryngology between September 2018 and April 2021. All participants provided signed informed consent at enrolment. The diagnosis of ENS was based on subjective symptoms, including paradoxical nasal obstruction, and comparable findings in a cotton test (during which a moistened cotton ball was placed in the widest area of the common nasal cavity). Improvement in symptoms through 30 min of nasal inhalation was indicative of a positive cotton test [9,16]. Clinical characteristics and demographic data, including body mass index (BMI), neck circumference (NC), and serum total immunoglobulin E (sIgE), were documented. Nasal endoscopy, bilateral nasal resistance measurement, and computed tomography were used to evaluate the nasal airway. Patients with congenital craniofacial anomalies or other sinonasal diseases, such as rhinosinusitis or nasal polyps, were excluded.

### 2.2. Total Nasal Resistance (TNR)

Nasal resistance for each nostril was measured by active anterior rhinomanometry at 150 Pa, after a rest period in the seated position. A pressure sensor was placed in one nostril, while airflow during inspiration was measured in the other. Unilateral nasal resistance values of the left (R_L_) and right (R_R_) nostrils were calculated according to Ohm’s Law (flow = pressure/resistance). TNR was obtained using the following formula:TNR = (R_L_ × R_R_)/(R_L_ + R_R_). 

### 2.3. Polysomnography

Each patient underwent full-night polysomnography (Alice 5, Respironics using standard techniques). Sleep stages and arousals were scored according to the American Academy of Sleep Medicine criteria [17]. Apnoea was defined as more than a 90% reduction in airflow for more than 10 s. Hypopnoea was defined as more than a 30% decrease in airflow for at least 10 s with associated oxygen desaturation of more than 3% or associated arousal. The apnoea–hypopnoea index (AHI), which refers to episodes of obstructive apnoea and/or hypopnoea per hour of sleep, and the lowest observed oxy-haemoglobin saturation by pulse oximetry (lowest SpO_2_) are the main indicators for assessment of OSA severity. Based on the polysomnography results, mild, moderate, and severe OSA were defined as AHI ≥ 5, 15, and 30 per hour, respectively.

### 2.4. Subjective Evaluation

The empty nose syndrome 6-item questionnaire (ENS6Q) is a validated, ENS-specific instrument to identify patients suspected of having ENS [18,19]. The ENS6Q consists of six items, including dryness, lack of air sensation going through the nasal cavities, suffocation, nose feeling too open, nasal crusting, and nasal burning. Participants scored each item from 0 (no symptom) to 5 (severe symptom), with a maximum score of 30 points. An ENS6Q score ≥ 10.5 suggests the possible presence of ENS to the clinician.

The Sino-Nasal Outcome Test-22 (SNOT-22) questionnaire is a validated instrument developed to evaluate subjective measures of symptoms in sinonasal disorders [20]. The SNOT-22 questionnaire includes various symptoms, physical problems, functional limitations, and emotional consequences. Scores range from 0 to 5, with higher scores indicating more severe symptoms.

The Epworth Sleepiness Scale (ESS) is the instrument most commonly used for assessing daytime sleepiness [21]. This self-administered questionnaire calculates eight scores about the frequency of sleepiness on a 4-point scale (0–3) in eight different situations in daily life. ESS scores >10 indicate extensive daytime sleepiness [22].

### 2.5. Statistical Analyses

Data are presented as mean ± standard deviation (SD) and were statistically analysed using SPSS version 26.0 (IBM Corp, Armonk, NY, USA). Univariate and multivariate linear regression models determined the association between AHI and clinical variables. Correlations were found using Pearson’s correlation coefficient (r) or Spearman’s correlation coefficient (r_s_) when data did not pass the normality test. To identify and characterise the sensitivity and specificity of clinical metrics for detection of moderate-to-severe OSA in ENS participants, receiver operating characteristic (ROC) curves were analysed and the area under the ROC curve (AUC) was calculated. Statistical significance was set at *p* < 0.05.

## 3. Results

### 3.1. Clinical Characteristics of the Study Population

A total of 48 ENS patients with a mean (±SD) age of 44.5 ± 11.7 years, including 7 women and 41 men, were enrolled during the study period. Table 1 summarises the general characteristics of the participants. The mean (±SD) BMI and neck circumference were 24.0 ± 3.1 kg/m^2^ and 37.3 ± 3.1 cm, respectively, which indicated that the patients were slightly overweight. The mean (±SD) scores of questionnaires, including ESS, SNOT-22, and ENS6Q, were 10.8 ± 6, 63.9 ± 18.1, and 15.2 ± 5.0, respectively, indicating significant symptomatic burdens on patients’ QOL.

### 3.2. PSG Results

Table 2 summarises the data collected from PSGs. The mean (±SD) AHI and the lowest SpO_2_ were 23.8 ± 22.4/h and 85.9 ± 11.1%, respectively, among 14 (29.2%), 12 (25.0%), and 14 (29.2%) patients with mild, moderate, and severe OSA. The mean (±SD) arousal index was 25.1 ± 15.9 per hour. In the sleep architecture analysis, stage N1, N2, N3, and rapid-eye-movement (REM) sleep comprised 30.2 ± 16.9%, 47.3 ± 15.5%, 2.1 ± 5.4%, and 20.0 ± 8.1% of the total sleep time. A relatively high proportion of stage N1 sleep and a low proportion of stage N3 sleep were observed in these patients when compared with the normal sleep architecture.

### 3.3. Regression Analyses for AHI

In the univariate regression analysis, BMI (β = 0.52, *p* < 0.001), NC (β = 0.52, *p* < 0.001), sIgE (β = 0.32, *p* = 0.026), and ENS6Q score (β = 0.31, *p* = 0.032) were significantly associated with AHI (Table 3). Further analysis using a multivariate regression model (adjusted r^2^ = 0.378, *p* < 0.001) revealed that BMI (β = 0.48, *p* < 0.001) and sIgE (β = 0.32, *p* = 0.01) were significant predictors of AHI.

### 3.4. Correlation Analyses of ENS6Q

The ENS6Q scores significantly correlated with AHI (r = 0.311, *p* = 0.032; Figure 1a), arousal index (r = 0.297, *p* = 0.040; Figure 1b), and ESS score (r = 0.623, *p* < 0.001; Figure 1c). The ENS6Q scores also negatively correlated with TNR (r_s_ = −0.482, *p* < 0.001; Figure 1d).

### 3.5. Using Clinical Metrics to Detect Moderate-to-Severe OSA

Given the concern that patients with moderate-to-severe OSA (AHI ≥ 15) warrant aggressive intervention and therapy, ROC curves were generated, and the AUC was calculated to evaluate whether any of the clinical variables would detect moderate-to-severe OSA in our participants better than a random test in a statistically significant manner (AUC > 0.5) (Figure 2). The ROC curves of BMI (AUC = 0.765, *p* < 0.001) and NC (AUC = 0.734, *p* < 0.001) were associated with AUCs significantly greater than 0.5. The optimal cut-offs for these variables (maximising the sum of sensitivity and specificity) were BMI > 25.3 kg/m^2^ (sensitivity: 61.5%, specificity: 95.5%) and NC > 38.5 cm (sensitivity: 57.7%, specificity: 95.2%).

## 4. Discussion

In adults, approximately 5%, 50%, and 20% of the total sleep time is stage N1, N2, and N3 sleep, respectively. The remaining 25% is REM sleep [23,24]. In the present study, stages N1, N2, N3, and REM sleep made up 30.2%, 47.3%, 2.1%, and 20.0% of the total sleep time, respectively. A marked increase in stage N1 and a decrease in stage N3 sleep were observed. Stage N1 sleep is the transition from an awake state to sleep. A high proportion of stage N1 sleep is generally a result of frequent arousal and sleep fragmentation caused by sleep disorders, such as sleep apnoea. Stage N3, sometimes referred to as slow-wave sleep, is considered to be deep sleep [25]. Our results confirmed substantial sleep dysfunction in ENS patients, which, in earlier subjective studies, was found to be associated with psychological burdens such as anxiety and depression [14,15]. Differentiating the associations between disease-specific impairment of QOL is important and could allow for targeted symptom therapy to improve patient-centred care in ENS because of the heterogeneity of these patients.

This is the first study to evaluate sleep architecture and investigate the prevalence of OSA by PSG in ENS patients. Earlier studies have reported that significant sleep dysfunction is associated with psychological burden in ENS patients [13,14,15]. The results of the present study showed a high prevalence of OSA in ENS patients; 40 (83.3%) patients had AHI ≥ 5 and 26 (54.2%) patients experienced moderate-to-severe OSA (AHI ≥ 15). In addition, AHI was significantly associated with BMI, NC, sIgE, and ENS6Q scores in patients with ENS. In detecting moderate-to-severe OSA by clinical metrics, BMI and NC were identified as significant predictors, with AUCs significantly greater than 0.5. The optimal cut-off values for these variables were BMI > 25.3 kg/m^2^ and NC > 38.5 cm. Moreover, ENS6Q scores were positively associated with AHI, arousal index, and ESS scores, and negatively correlated with TNR. These results show that low TNR and severe ENS symptoms may lead to significant sleep dysfunction and arousal. The extent of OSA was associated with the degree of obesity and severity of ENS symptoms in these patients.

The association of ENS and OSA in our patient group may be attributed to aggressive manipulation of the inferior turbinates in OSA patients who undergo surgery for nasal obstruction, or to improve the use of nasal continuous positive airway pressure [26]. An earlier study reported that oropharyngeal surgery for OSA can lower TNR in patients with moderate-to-severe OSA [27]. This indicates that TNR may be caused by both nasal and oropharyngeal stenosis in patients with moderate-to-severe OSA. Thus, there is a higher chance of over or repeated resection of the turbinates in these patients if the surgeon does not evaluate their upper airway comprehensively.

Previous studies by computational fluid dynamics (CFD) demonstrated that total turbinectomy reduced nasal resistance, increased nasal airflow rate [28], and increased airflow distribution around the middle meatus [29]. Thus, another possible mechanism for the association between ENS and OSA is that the reduced TNR and increased rate of airflow after nasal surgery may decrease the supporting pressure and increase the collapsibility of the narrow oropharynges in OSA patients, according to Bernoulli’s law. However, further CFD studies on the relationship between nasal airflow and oropharyngeal flow, resistance, and collapsibility in OSA patients are required.

Our previous study also reported significant psychological burdens in ENS patients, including that 38.3% and 53% of ENS patients experienced moderate-to-severe depression and anxiety, respectively [14]. The association between OSA and depression has been previously reported [30,31,32]. For example, in subjects with OSA, the prevalence of depression may reach 63% [31], whereas the prevalence of OSA in individuals with major depression was 36.3% [32]. The prevalence of OSA in the general population is 1–14% (2–7% of women and 9–14% of men) [33]. Hence, OSA seems to be more common in individuals with depression than in the general population. The high prevalence of obesity and metabolic syndrome associated with a depressive disorder may be a possible aetiology of OSA [34]. However, increased alcohol consumption, smoking, and benzodiazepine drug use in patients with depression and anxiety also potentially promote the development of OSA because these factors may cause a decrease in muscle tone or an increase in mucus secretion in the upper airway [35]. This could be another explanation for the high prevalence of OSA in ENS patients.

In addition to excessive daytime sleepiness, moderate-to-severe OSA usually leads to significant impairments in QOL and cognitive performance, and to an increase in traffic or occupational accidents [36]. In addition, moderate-to-severe OSA is significantly associated with cardiovascular and metabolic diseases, such as systemic arterial hypertension, coronary artery disease, heart failure, and stroke. All these factors could lead to substantial morbidity and mortality [37,38]. As a result, early identification and treatment of patients with moderate-to-severe OSA are important. In a study of the general population, male sex, age ≥ 50 years, NC ≥ 14.5 inches (36.8 cm), and BMI ≥ 30 kg/m^2^ are risk factors for moderate-to-severe OSA [39]. In this study, the ROC curve analysis revealed that BMI > 25.3 kg/m^2^ and NC >38.5 cm were predictors of moderate-to-severe OSA. Recognition of individuals experiencing marked OSA and provision of appropriate intervention are also critical to prevent long-term morbidity and mortality in ENS patients.

Our results also showed an association between sIgE levels and AHI. The relationship between allergies and OSA remains controversial. Allergic rhinitis-related inflammation in the upper airway might worsen the extent of OSA, and treatment of allergic rhinitis may reduce the severity of OSA-impaired sleep quality and daytime sleepiness [40]. However, a previous study showed no significant difference in the frequency of allergic rhinitis and atopy between subjects with and without OSA [41]. Future studies on the relationship between allergic inflammation and OSA severity in patients with ENS are necessary.

A limitation of this study is the cross-sectional evaluation of the association between variables. The aetiology of OSA is usually multifactorial [1]. ENS6Q is one of the frequently used symptom scores in the evaluation of ENS. We found an association between AHI and ENS6Q with correlation and univariate regression analysis, but not with multivariate regression; this may be due to the strong influencing factor of obesity in OSA [22]. Future studies that longitudinally analyse the metrics before and after treatment, such as surgery, as well as comparison with a control group, may be beneficial in illustrating the interaction of these variables. This study reported a high prevalence of OSA and significant sleep dysfunction in patients with ENS.

## 5. Conclusions

We found a high prevalence of OSA and significant sleep impairment in patients with ENS. The extent of OSA was associated with the degree of obesity and severity of ENS symptoms in these patients. In addition, the ENS6Q scores were negatively correlated with nasal resistance and positively associated with arousal frequency and daytime sleepiness. Recognition of individuals experiencing marked OSA and provision of appropriate intervention are critical to preventing long-term morbidity and mortality and improving therapeutic outcomes in ENS patients.

## Figures and Tables

**Figure 1 diagnostics-12-01720-f001:**
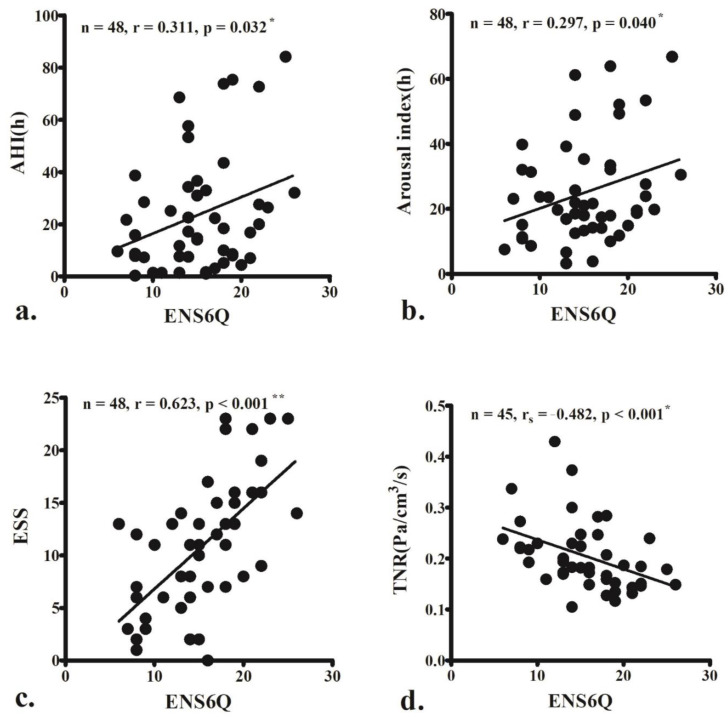
The empty nose syndrome 6-item questionnaire (ENS6Q) scores are significantly associated with the apnoea–hypopnoea index (AHI) (**a**), arousal index (**b**), and Epworth Sleepiness Scale (ESS) score (**c**). ENS6Q is also negatively correlated with total nasal resistance (TNR) (**d**). The correlations were determined using Pearson’s correlation coefficient (r) or Spearman’s correlation coefficient (r_s_) if the data did not pass the normality test. (r, 0–0.2: very weak, 0.2–0.4: weak, 0.4–0.6: moderate, 0.6–0.8: strong, 0.8–1.0: very strong correlation) * *p* < 0.05, ** *p* < 0.001.

**Figure 2 diagnostics-12-01720-f002:**
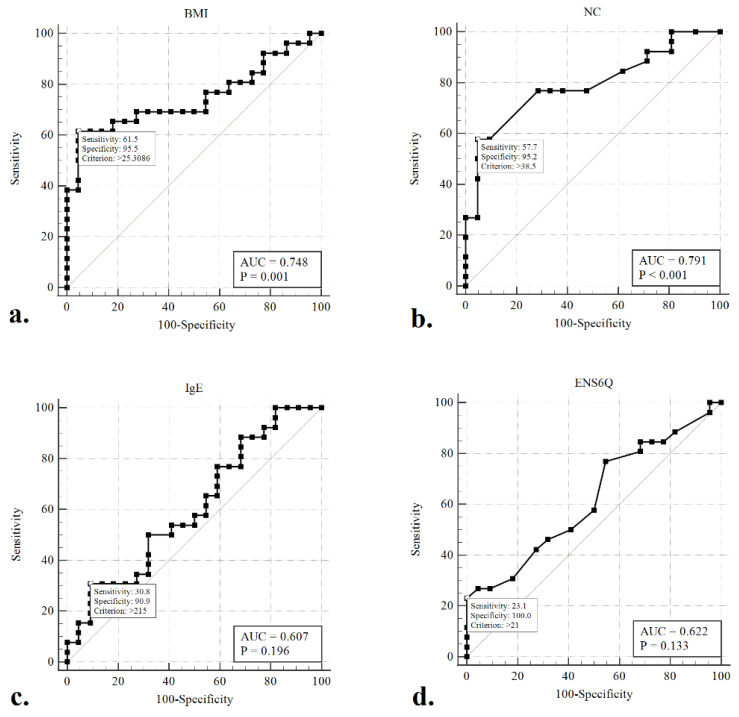
Receiver operating characteristic (ROC) curves to detect moderate-to-severe obstructive sleep apneoa (AHI > 15) using the variables of BMI (**a**), NC (**b**), sIgE (**c**), and ENS6Q (**d**). The optimal cut-offs for these metrics (maximizing the sum of sensitivity and specificity) are indicated.

**Table 1 diagnostics-12-01720-t001:** Clinical characteristics of the study participants.

Variables	Mean ± SD	Range
Age (year)	44.5 ± 11.7	24–67
Female:male, n	7:41	
Smoking, n	11	
BMI (kg/m^2^)	24.0 ± 3.1	17.5–31.6
NC (cm)	37.3 ± 3.1	31–44
sIgE (IU/mL)	278.6 ± 546.1	2–2403
TNR (Pa/cm^3^/s)	0.20 ± 0.07	0.11–0.43
ESS	10.8 ± 6.1	1–23
SNOT-22	63.9 ± 18.1	33–108
ENS6Q	15.2 ± 5.0	6–26

BMI, body mass index; NC, neck circumference; sIgE, serum total immunoglobulin E; TNR, total nasal resistance; ESS, Epworth Sleepiness Scale; SNOT-22, Sino-Nasal Outcome Test-22; ENS6Q, empty nose syndrome 6-item questionnaire.

**Table 2 diagnostics-12-01720-t002:** Polysomnographic results of the study participants.

Variables	Mean ± SD	Range
AHI (/h)	23.8 ± 22.4	0.3–84.2
Mild OSA, n (%)	14 (29.2)	
Moderate OSA, n (%)	12 (25.0)	
Severe OSA, n (%)	14 (29.2)	
Arousal Index (/h)	25.1 ± 15.9	3.2–66.8
Snore index (/h)	266.5 ± 162.9	2.6–593.6
Lowest SpO_2_ (%)	85.9 ± 11.1	45–98
Mean SpO_2_ (%)	96.6 ± 1.8	91–99
Efficiency (%)	73.4 ± 15.3	25.8–96.1
Stage A (%)	20.9 ± 13.8	3.5–73.5
Stage REM (%)	20.0 ± 8.1	2.2–36.2
Stage N1 (%)	30.2 ± 16.9	6.7–78.6
Stage N2 (%)	47.3 ± 15.5	5.0–75.5
Stage N3 (%)	2.1 ± 5.4	0.0–27.1

AHI, apneoa–hypopneoa index; OSA, obstructive sleep apneoa; lowest SpO_2_, lowest oxyhaemoglobin saturation by pulse oximetry; A, awake; REM, rapid eye movement.

**Table 3 diagnostics-12-01720-t003:** Linear regression analysis for an apnoea–hypopnoea index of the study participants.

	Univariate Analysis			Multivariate Analysis			
Variables	B (95% CI)	β	*p*	B (95% CI)	β	*p*	VIF
Age(year)	0.44 (−0.11–0.99)	0.23	0.115				
BMI(kg/m^2^)	3.71 (1.19–5.52)	0.52	<0.001 **	3.44 (1.58–5.30)	0.48	<0.001 **	1.166
NC(cm)	3.83 (1.96–5.69)	0.52	<0.001 **	0.23 (−0.66–1.12)	0.03	0.601	1.036
sIgE(IU/mL)	0.01 (0.002–0.025)	0.32	0.026 *	0.01 (0.003–0.023)	0.32	0.01 *	1.032
TNR(Pa/cm^3^/s)	−44.61 (−148.08–58.85)	−0.13	0.389				
ESS	0.98 (−0.06–2.02)	0.27	0.065	0.07 (−1.10–1.25)	0.02	0.903	1.790
SNOT-22	0.34 (−0.02–0.69)	0.27	0.063	−0.12 (−0.53–0.29)	−0.10	0.547	1.879
ENS6Q	1.40 (0.13–2.67)	0.31	0.032 *	1.13 (−0.41–2.68)	0.25	0.147	2.042

BMI, body mass index; NC, neck circumference; sIgE, serum total immunoglobulin E; TNR, total nasal resistance; ESS, Epworth Sleepiness Scale; SNOT-22, Sino-Nasal Outcome Test-22; ENS6Q, empty nose syndrome 6-item questionnaire; VIF, variance inflation factor. * *p* < 0.05, ** *p* < 0.001.

## Data Availability

The dataset used and/or analysed during the current study is available from the corresponding author upon reasonable request.

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
