# Peer review of "Identifying Obstructive Sleep Apnoea in Patients with Empty Nose Syndrome"

_diagnostics, 2022, doi:10.3390/diagnostics12071720_

Round 1

Reviewer 1 Report

The paper is well written and solid. A high prevalence of OSA and significant sleep impairment in patients with ENS was found. I would recommend it for publication if the following minor issues can be addressed.

1. Enlarge panels in Figure 1 so that it becomes really visible to readers.

2. Increase the fonts size in Figure 2.

Author Response

 The paper is well written and solid. A high prevalence of OSA and significant sleep impairment in patients with ENS was found. I would recommend it for publication if the following minor issues can be addressed.

  1. Enlarge panels in Figure 1 so that it becomes really visible to readers.

Reply:

We appreciated the suggestion of the reviewer. We enlarged the panels in Figure 1.

  1. Increase the fonts size in Figure 2.

Reply:

We appreciated the suggestion of the reviewer. We increased the fonts size in Figure 2.

Reviewer 2 Report

This study was to investigate “Identifying obstructive sleep apnoea in patients with empty nose syndrome”. The authors found a higher prevalence of OSAS in patients with empty nose syndrome.

There are several concerns in this manuscript.

1.               As the authors mentioned, there is lack of a control group to compare with patients with empty nose syndrome. Therefore, it is difficult to decide whther the high prevalence of OSAS in patients with empty nose syndrome is caused by empty nose syndrome itself or other co-exsitenent factors, for example obsesity or nasal obstruction.

2.               As the multivariate regression model showed ENS6Q was not a predictor of AHI, there was no evidence to support that empty nose syndrome was related to OSAS.

3.               Furthermore, the ROC analysis did not show empty nose syndrome was related to mod.-severe OSAS.

Author Response

We appreciate the valuable comments, and have responded to all their comments point-by-point. We have also made appropriate changes to the manuscript as requested by the reviewers. We enclose a revised manuscript which incorporates these changes. 

  1. As the authors mentioned, there is lack of a control group to compare with patients with empty nose syndrome. Therefore, it is difficult to decide whther the high prevalence of OSAS in patients with empty nose syndrome is caused by empty nose syndrome itself or other co-exsitenent factors, for example obsesity or nasal obstruction.

Reply:

We appreciated the comment of the reviewer. We agreed with it.

Future studies with longitudinally analyse the metrics before and after treatment, such as surgery, as well as comparison with a control group may be beneficial in illustrating the interaction of these variables. This study reported a high prevalence of OSA and significant sleep dysfunction in patients with ENS. (Page 9, Discussion)

  1. As the multivariate regression model showed ENS6Q was not a predictor of AHI, there was no evidence to support that empty nose syndrome was related to OSAS.

Reply:

We appreciated the comment of the reviewer.  

In the univariate regression analysis, BMI (β = 0.52, p < 0.001), NC (β = 0.52, p < 0.001), sIgE (β = 0.32, p = 0.026), and ENS6Q score (β = 0.31, p = 0.032) were significantly associated with AHI. Further analysis using a multivariate regression model (adjusted r2 = 0.378, p < 0.001) revealed that BMI (β = 0.48, p < 0.001) and sIgE (β = 0.32, p = 0.01) were significant predictors of AHI.

The etiology of OSAS is usually multifactorial. ENS6Q is one of the frequently used symptom score in evaluation of ENS. We found an association between AHI and ENS6Q with correlation and univariate regression analysis, but not with multivariate regression. It may be due to the strong influencing factor of obesity. Further study with investigation of the AHI change after ENS reconstruction is ongoing and would be better in answering these issue. This study reported a high prevalence of OSA and significant sleep dysfunction in patients with ENS. (Page 9, Discussion)

  1. Furthermore, the ROC analysis did not show empty nose syndrome was related to mod.-severe OSAS.

Reply:

We appreciated the comment of the reviewer.

Given the concern that patients with moderate-to-severe OSA (AHI ≥ 15, n=26) warrant aggressive intervention and therapy, ROC curves were generated, and the AUC was calculated to evaluate whether any of the clinical variables would detect moderate-to-severe OSA in our participants better than a random test in a statistically significant manner (AUC > 0.5). The ROC curves of BMI (AUC = 0.765, p < 0.001) and NC (AUC = 0.734, p < 0.001) with AUCs significantly greater than 0.5.

These results indicate obesity plays the crucial role in moderate-to-severe OSA. Further study with investigation of the AHI change after ENS reconstruction is ongoing and would be better in answering these issue. This study reported a high prevalence of OSA and significant sleep dysfunction in patients with ENS.

We hope our reply is satisfactory and we look forward to hearing from you in due course.

Yours sincerely

Ta-Jen Lee, MD

Round 2

Reviewer 2 Report

I do not have further comments.